

# An Effective Communication Topology for Performance Optimization: A Case Study of the Finite Volume WAve Modeling (FVWAM)

Renbo Pang[1,2], Fujiang Yu[1,2], Yuanyong Gao[1], Ye Yuan[1,2], Liang Yuan[3], and Zhiyi Gao[1]

[1]  National Marine Environmental Forecasting Center, 8 Dahuisi Road, Beijing, 100080, China,

[2]  Key Laboratory of Research on Marine Hazards Forecasting, Ministry of Natural Resources of China, 8 Dahuisi Road, Beijing, 100080, China

[3]  Institute of Computing Technology, Chinese Academy of Sciences, 6 Kexueyuan Nanlu, Zhongguancun, Haidian, Beijing, 100190, China

*Correspondence to*: Renbo Pang (pangrb@nmefc.cn), Fujiang Yu (yvfujiang_2022@163.com), Yuanyong Gao (gaoyy@nmefc.cn)

**Abstract.** High-resolution models are essential for simulating small-scale processes and topographical features, which play
a crucial role in understanding meteorological and oceanic events, as well as climatic patterns. High-resolution modeling requires substantial improvement on the parallel scalability of the model to reduce runtime, while massive parallelism is associated with intensive communications. Point-to-point communication is extensively utilized for neighborhood communication in earth models due to its flexibility. The distributed graph topology, first introduced in the MPI version 2.2, provides a scalable and informative communication method. It has demonstrated significant speedups over the point-to-point communication
method based on a variety of synthetic and real-world communication graph datasets. But its application in earth models for neighborhood communication is rarely studied. In this study, we implemented neighborhood communication using both the traditional point-to-point communication method and the distributed graph communication topology. We then compared their performance in a case study of the Finite Volume WAve Modeling (FVWAM). Across all tests with 512 to 32,768 processes, the communication time speedup of the distributed graph communication topology ranged from 1.28 to 5.63 compared to the
point-to-point communication method. For operational global wave forecasts with 1,024 processes, the runtime of the FVWAM reduced 40.2% when the point-to-point communication method was replaced by the distributed graph communication topology.

## 1  Introduction

Numerical earth models with higher resolution are capable of more accurately representing small-scale processes and topo-
20 graphical features, which are essential for phenomena of weather and sea, and finer details of the climate (Palmer, 2019). For instance, fine meshes with 1/10° resolution or better are needed to simulate emerging eddy dynamics (Koldunov et al., 2019). Submesoscale eddies are believed to affect mixed layer restratification and vertical heat transport (Su et al., 2018). These



eddies may also contribute to shaping the circulation in major current systems (Chassignet and Xu, 2017). Additionally, A high-resolution spectral wave model can deal with shallow water conditions and incorporate the interaction due to tide and surge (Monbaliu et al., 2000).

The performance of top computing clusters has been increasingly improved with the development of semiconductors and the emergence of hybrid computing systems accelerated by General-Purpose Graphics Processing Units (GPGPUs). For instance, the first high performance computer (HPC) with over 1 exaflop/s computing performance was established at the Oak Ridge National Laboratory of the United States in 2022 (Sukhija et al., 2022). These high performance computing systems provide the necessary resources to run models with higher resolutions. Gu et al. (2022) developed the integrated Atmospheric Model Across Scales (iAMAS) with a global 3km spatial resolution. Wedi et al. (2020) evaluated a 4-month global atmospheric simulation with ECMWF's (European Centre for Medium-Range Weather Forecasts) hydrostatic Integrated Forecasting System (IFS) at an average grid spacing of 1.4km. Zhang et al. (2023) used the coupled Earth system model (ESM) to simulate sea and ice with a global 3km resolution.

However, there are many challenges to implement and apply these models with higher resolutions (Alizadeh, 2022). One critical factor hampering the performance of these high-resolution models is their limited parallel scalability (Koldunov et al., 2019). For instance, the barotropic solver is a major bottleneck in the Parallel Ocean Model (POP) within the high-resolution CESM, which scales poorly at high process counts due to inherent communication limitations in the algorithm (Huang et al., 2016). The existing models struggle to make full use of the new generation of massively parallel HPC systems (Koldunov et al., 2019). In parallel computing, data exchange introduces additional costs compared to serial computing. As the number of parallel processes increases, time of computation in each process theoretically decreases in proportion. However, time of communication in each process is reduced at a slower rate compared to computation or even increases with more processes. The most common communication in atmospheric and oceanic models is neighborhood communication to exchange data of local grids in each process with data of neighboring grids in other processes (Wolters, 1992). How to reduce neighborhood communication cost is a key factor in improving the parallel scalability of models (Ovcharenko et al., 2012).

The point-to-point communication interfaces implemented by send/receive routines in the Message-Passing Interface (MPI) standard version 1.0 are basic and flexible methods for data exchange (Walker and Dongarra, 1996). They are extensively utilized for neighborhood communication in atmospheric and oceanic models including the Weather Research and Forecasting model (WRF) (Biswas et al., 2018), the Model for Prediction Across Scales (MPAS) (Sinkovits and Duda, 2016), Nucleus for European Modelling of the Ocean (NEMO) (Irrmann et al., 2021), IFS (Mozdzynski et al., 2015), WAve Modeling (WAM) (Katsafados et al., 2016), Atmospheric General Circulation Models (AGCMs) (Wang et al., 2017), etc. A newly scalable and informative communication method of the distributed graph topology was provided in the MPI standard version 2.2 (Hoefler et al., 2011). This topology is capable of generating a new communicator that reorders processes to better match the capabilities of the underlying hardware (Mirsadeghi et al., 2017). Ghosh et al. (2019) demonstrated speedups of 1.4 to 6 times using the distributed graph topology (employing up to 16,000 processes) compared to the point-to-point communication method. They evaluated this approach using a variety of synthetic and real-world communication graph datasets, including random geometric



graphs, graph500 R-MAT, stochastic block partitioned graphs, protein k-mer, DNA, CFD, and social networks. To date, the application of this topology in earth models for communication optimization has been infrequent.

Ocean wave modeling holds significant importance within numerical weather prediction systems, not only for its crucial role in ship routing and offshore engineering but also due to its climate implications (Yuan et al., 2024). Recently, the National Marine Environmental Forecasting Center of China (NMEFC) developed the Finite Volume WAve Modeling (FVWAM) based on the WAM for national operational wave forecasting services. The FVWAM employs a neighboring communication pattern based on spatially decomposed grid results among multiple processes, a common approach in oceanic and atmospheric models.

Consequently, we implemented neighborhood communication using both the point-to-point communication method and the distributed graph communication topology in a case study of the FVWAM based on Spherical Centroidal Voronoi Tessellation (SCVT) grids. The contribution of this paper includes: 1) to the best of the authors' knowledge, the first application of the distributed graph communication topology in a global wave model for neighborhood communication; 2) verifying that the distributed graph topology achieves superior communication performance over the point-to-point communication method in the case study of the FVWAM; 3) providing a method for optimizing neighborhood communication in earth models based on spatial decomposition including both unstructured grids and structured grids.

The remainder of this paper is organized as follows: Section 2 summarizes the related work, including descriptions of the FVWAM and the distributed graph communication topology. Section 3 introduces the design of the distributed graph communication topology and the point-to-point communication method. Section 4 presents experimental results, evaluations of two communication methods, and products of the FVWAM. Finally, the paper concludes in Section 6.

## 2 Description

### 2.1 Description of the FVWAM

Ocean waves can be considered as a combination of wave components across a frequency and direction spectrum in geographic space and time. Their generation, propagation, dissipation, and nonlinear interaction processes are described by a wave action transport equation with wave propagation and source terms placed on both sides of the equation. A series of spectral wave models have been developed by numerically solving this wave action transport equation, including the WAM (Group, 1988), WaveWatchIII (Tolman et al., 2009), and the SWAN (Booij et al., 1999).

Based on the WAM model, the FVWAM developed by the NMEFC is a third-generation spectral wave model based on the wave action equation 1. In this equation, $N$ stands for wave action, $t$ denotes time, $X$ stands for spatial coordinates, $\theta$ represents direction and $\sigma$ denotes angular frequency. The left-hand side of Equation 1 accounts for the spatial and intra-spectral propagation of spectral energy, respectively representing the change of wave action in time, the propagation of wave action in spatial coordinates, the propagation velocities in spectral space. The source term ($S_{tot}$) on the right-hand side of Equation 1 includes wind input, dissipation due to whitecapping, bottom friction, depth-induced wave breaking, and the exchange of wave action between spectral components due to nonlinear effects.



$$\frac{\partial N}{\partial t} + \nabla_{\boldsymbol{X}}(\dot{\boldsymbol{X}}N) + \frac{\partial}{\partial\theta}(\dot{\theta}N) + \frac{\partial}{\partial\sigma}(\dot{\sigma}N) = \frac{S_{tot}}{\sigma} \tag{1}$$

Compared to the WAM, the FVWAM replaces spherical latitude-longitude grids with SCVT grids to better accommodate coastal topographic features. For advection in spectral and directional space, the FVWAM employs the same second-order central differencing method as the WAM. Regarding the integration of source terms, the FVWAM implements the same semi-implicit integration scheme as adopted in the WAM. The detailed description of the advection and source terms in the WAM can be referred to the WAMDI GROUP publication (Group, 1988).

Spatial decomposition is employed to partition data in the FVWAM for parallelization. This approach involves dividing computational tasks across multiple processors or nodes. Data exchange is necessary among neighboring processes for wave action ($N$), water depth, and bathymetric gradient in the FVWAM. The most time-consuming communication in the FVWAM occurs during the exchange of wave action for the integration of advection. $N$ is the only 3D variable in the FVWAM, whose dimension size is the total grid count, number of angular frequencies and directions. It is exchanged once in each integration time step. Therefore, the exchange of $N$ is utilized as a benchmark for evaluating the performance of neighboring communication in the case study of FVWAM.

## 2.2   Description of the distributed graph communication topology

The distributed graph communication topology in MPI supports optimizing communication by minimizing communication costs and enhancing load balance (Mozdzynski et al., 2015) (Traff, 2002). This topology mechanism leverages user-provided topology information to reorder processes within a new communicator, aligning them more effectively with the underlying network to achieve higher communication performance (Mirsadeghi et al., 2017). The impact of process ordering on communication performance is illustrated in Figure 1 and Figure 2. Figure 1 depicts a simple neighborhood communication pattern among computing processes. The squares represent the processes, the numerals within the squares indicate process IDs (ranging from 0 to 5), and the dashed arrows signify the communication links between pairs of processes. For instance, Process 0 is engaged in data exchange with its neighboring Process 1 and Process 3.

    Based on the neighborhood communication pattern in Figure 1, assigning two processes to each node can lead to several process mapping results among computing nodes, as shown in Figure 2. The squares represent computing nodes, and the numerals within the squares denote the process IDs. There are four bi-directional communication links among three computing nodes in Figure 2(a) and Figure 2(b), and seven bi-directional communication links among three computing nodes in Figure 2(c). The process mapping topology in Figure 2(c) is the least efficient and most time-consuming among the presented mapping results. It exists a communication hotspot with four communication links in the node housing Process 1 and Process 4 in Figure 2(b). The node containing Process 3 and Process 4 in Figure 2(a) has a maximum of three communication links, thus the process mapping result in Figure 2(a) is the most effective. These differing process mapping results can influence communication performance and the parallel scalability of applications. The distributed graph communication topology in MPI is capable of optimizing the mapping of computing processes in accordance with the underlying network and the communication pattern specified by users.





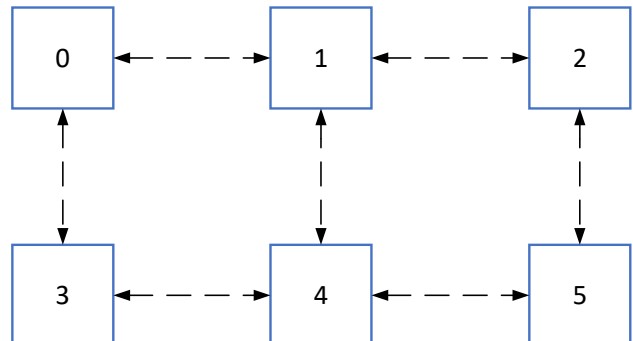

**Figure 1.** An example of neighborhood communication pattern

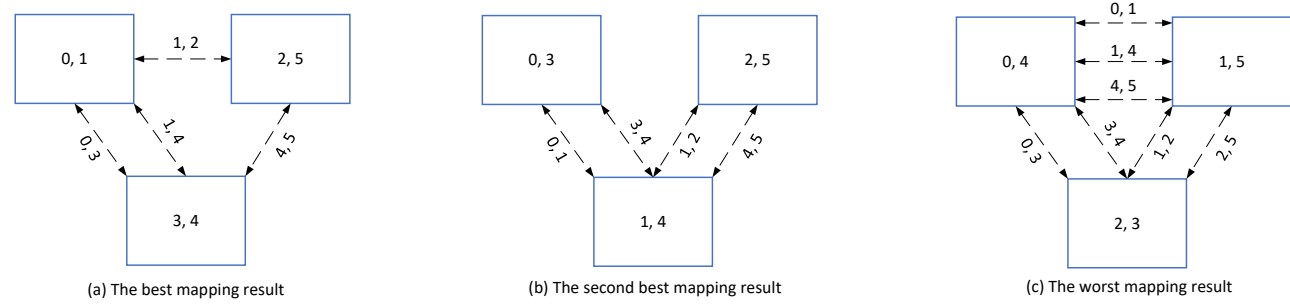

**Figure 2.** Process mapping results

This is the advantage for the distributed graph communication topology to achieve high performance. However, the trade-off associated with the distributed graph communication topology is the necessity to create the graph communication topology and allocate all communication data buffers before commencing communication. In contrast to the point-to-point communication method, users can neither specify the order of neighboring processes to send/receive data, nor they can reuse the same data buffer for multiple send/receive data with different neighboring processes in the distributed graph communication topology.

## 3 Design

### 3.1 Distributed graph communication topology

The workflow to create a distributed graph communication topology based on SCVT grids is delineated in Figure 3. Initially, the global SCVT grids are decomposed by partitioning tools in accordance with the number of computing processes. A simple partitioning result of the global SCVT sea grids into three partitions is shown in Figure 3(a). The three partitions are separately colored in green, blue, and purple. The partitioning result includes the mapping relationship between each grid ID and its corresponding process ID.





Subsequently, each process ascertains its receiving process IDs and grid IDs based on the partitioning result in Figure 3(a) and neighboring grid and process information in Figure 3(b). The variable of *CellsonCell* in the SCVT grid file discloses neighboring grid IDs for each grid. For instance, the neighboring grid IDs for the grid cell $C_1$ are the grid cells ($C_{2,3,4,8,9,11}$). The green grid cells ($C_{1,2,3,4}$) are assigned to Process $P_0$, while the blue grid cells ($C_{8,9,11}$) are assigned to Process $P_1$. The red line denotes the boundary separating the grid cells allocated to Processes $P_0$ and $P_1$, while the orange line delineates the communication boundary for Process $P_0$. The grid cells situated between the red local grid boundary line and the orange communication boundary line comprise the receiving grid cells for Process $P_0$. We can derive all receiving grid IDs by identifying all neighboring grid IDs of each process's local grids and then excluding the local grid IDs from these neighboring grid IDs. Utilizing the mapping relationship between each grid ID and its responding process ID, the receiving process IDs can be inferred from the receiving grid IDs.

Thirdly, the sending process IDs and the sending grid IDs can be deduced from the receiving process IDs and the receiving grid IDs conforming to the reciprocity inherent to paired sending and receiving operations. A distributed graph communication topology can be established by calling the MPI interface with reference to the sending and receiving process IDs and their associated degrees in Figure 3(c). The sending degree represents the total count of sending process IDs, and the receiving degree denotes the total count of receiving process IDs. The sending and receiving grid IDs are used as indices for exchanging data in the distributed graph communication topology.

The method to create a distributed graph communication topology and implement data exchange is delineated in Figure 4. In step 1, a multilevel partitioning scheme provided by METIS (Karypis and Kumar, 1997) is employed for the partitioning of the global SCVT grids. The METIS tool is compatible with both structured and unstructured grids and has been extensively utilized in various models, including MPAS (Heinzeller et al., 2016), WAVE WATCH III (WW3) (Abdolali et al., 2020), Finite Volume Coastal Ocean Model (FVCOM) (Cowles, 2008), etc. For the utilization of METIS, it is requisite to provide the total number of grids, the total number of edges between two neighboring grids, and neighboring grid IDs for each grid as inputs. The total number of grids is represented by the scalar variable of *nCells*, the total number of edges is represented by the scalar variable of *nEdges*, and the neighboring grid IDs are denoted by the two-dimensional variable of *cellsOnCell*, excluding invalid edges present in the SCVT grid file.

In step 2, the method to search receiving grid IDs is executed within a nested two-level loop structure. The external loop is sequential to traverse all local grid IDs from the lowest to highest value. The internal loop executes a sequential search for all neighboring grid IDs of each local grid ID using *cellsOnCell*($c, i$). $c$ represents the current grid ID, $i$ indicates the index of edges for the grid ID $c$, and *cellsOnCell*($c, i$) denotes the neighboring grid IDs of the grid ID $c$. If the neighboring process ID differs from the current process ID, then *cellsOnCell*($c,i$) is identified as one of receiving grid IDs for the current process ID. Based on all receiving grid IDs, the receiving process IDs can be ascertained through the mapping relationship between each grid ID and its corresponding process ID, which is determined in Step 1.

The order of receiving grids becomes disordered after Step 2. These grids are sorted in Step 3 to ensure the continuity of data exchange with neighboring processes and to enhance the cache hit rate for improving performance. The sorting procedure must fulfill two criteria. First, the receiving grid IDs from the same receiving process ID should be arranged continuously and



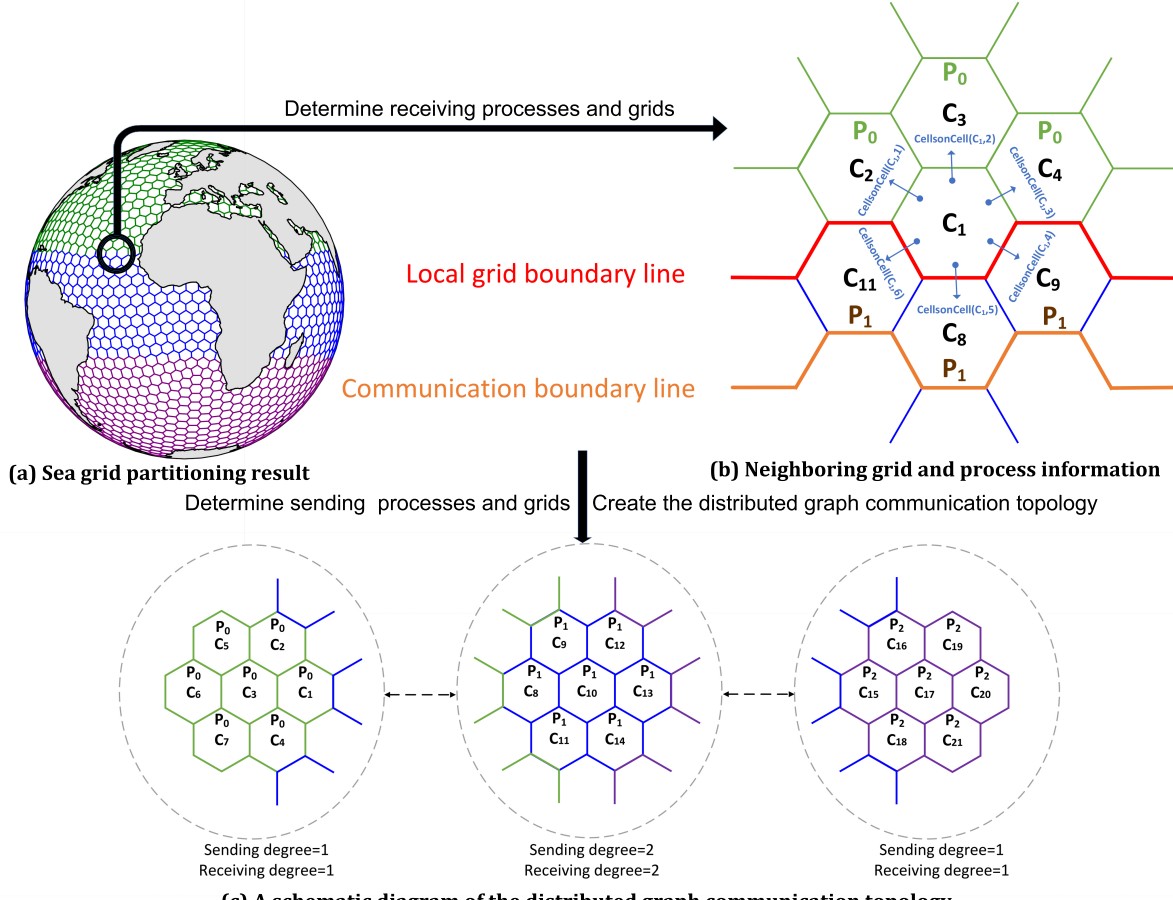

**Figure 3.** The workflow to create a distributed graph communication topology

sorted in ascending order based on the receiving process IDs. Second, the receiving grid IDs from the same process ID need to be sequenced from the lowest to the highest value. The quicksort method is employed to sort the receiving grid IDs that are located in the same receiving process ID.

In step 4, sending grid IDs and sending process IDs are ascertained by the primary process $P_0$. This is achieved by collecting the receiving grid IDs and receiving process IDs from the other processes, and subsequently disseminating the results to all processes. Initially, each process sends its receiving process IDs and receiving grid IDs to the process $P_0$. Subsequently, the process $P_0$ determines the sending process IDs using the rationale that the process which sends the receiving process ID to the process $P_0$ is the sending process ID for this process. Because the receiving process IDs have been sorted in Step 3 and the searching process is sequential, we only need to store the identified process IDs in sequence without the requirement for further sorting. Third, the process $P_0$ scatters these sending process IDs to the corresponding processes. Finally, the sending grid IDs





within the sending process are identical to the receiving grid IDs in the receiving process. The procedure to collect and scatter sending grid IDs is analogous to that used for sending process IDs.

In Step 5, based on the receiving process IDs in Step 2 and the sending process IDs in Step 4, the distributed graph communication topology is created by calling the MPI interface of MPI_DIST_GRAPH_CREATE_ADJACENT(*sources*, *destinations*, *reorder*, *comm_dist_graph*, ...). *sources* represents the array of receiving process IDs, *destinations* denotes the array of sending process IDs. The parameter of *comm_dist_graph* represents the new communicator endowed with the distributed graph topology, which is subsequently used for neighboring communication. The parameter of *reorder* is of boolean type. When it is true,

this interface reorders the process IDs within *comm_dist_graph* for optimizing communication based on receiving process IDs, sending process IDs, and network hardware structure. When it is false, the process IDs in *comm_dist_graph* preserve the same process order before creating the distributed graph communication topology.

     In step 6, neighboring data exchange is executed by calling the MPI distributed graph communication interface MPI_-NEIGHBOR_ALLTOALLV(*sendbuf*, *recvbuf*, *comm_dist_graph*, ...). The parameter *sendbuf* denotes the data buffer corre-

190 sponding to the sending grid IDs in Step 4, and *recvbuf* represents the data buffer corresponding to the receiving grid IDs in Step 3. The parameter *comm_dist_graph* is the communicator created in Step 5. This interface completes all the sending and receiving operations in a single function call. Data sent to different processes are stored contiguously in *sendbuf* following the order of sending processes as listed in *destinations* in Step 5. Similarly, data received from different processes are stored contiguously in *recvbuf* according to the order of receiving processes as specified in *sources* in Step 5. Compared to sending

and receiving operations implemented by users through calling multiple send/receive MPI interfaces, the MPI distributed graph communication topology is more user-friendly and efficient, significantly reducing the risk of deadlocks that could arise from improper use of multiple send/receive operations.

### 3.2    Point-to-point communication method

     A common approach to using the point-to-point communication method for neighboring communication is depicted in Figure 5.

The procedure to ascertain ordered arrays of receiving grid IDs, receiving process IDs, sending grid IDs, and sending process IDs, is the same as Step 1-4 of the distributed graph communication topology presented in Figure 4.

     To avoid communication deadlocks, the FVWAM initiates non-blocking receiving operations before commencing sending operations in Step 5 as the MPAS (Heinzeller et al., 2016) and the NEMO (Epicoco et al., 2011). Each process executes receiving operations to receive data from the corresponding receiving processes. These receiving operations are implemented

by repeatedly calling the MPI interface MPI_irecv (*recvbuf*, *source*,...). The parameter of *recvbuf* represents the data buffer associated with the receiving grid IDs from a single receiving process, and the parameter of *source* denotes the single receiving process ID. The frequency of calling the MPI_irecv interface is determined by the count of receiving process IDs for each process. This interface returns immediately and does not necessitate waiting for the completion of the receiving operation.

     In step 6, each process calls the MPI interface mpi_send (*sendbuf*, *destination*) to send data to the sending processes. The

210 parameter of *sendbuf* is designated for storing data corresponding to the sending grid IDs, and the parameter of *destination* indicates a single sending process ID. The mpi_send is a blocking communication interface that concludes after the sending




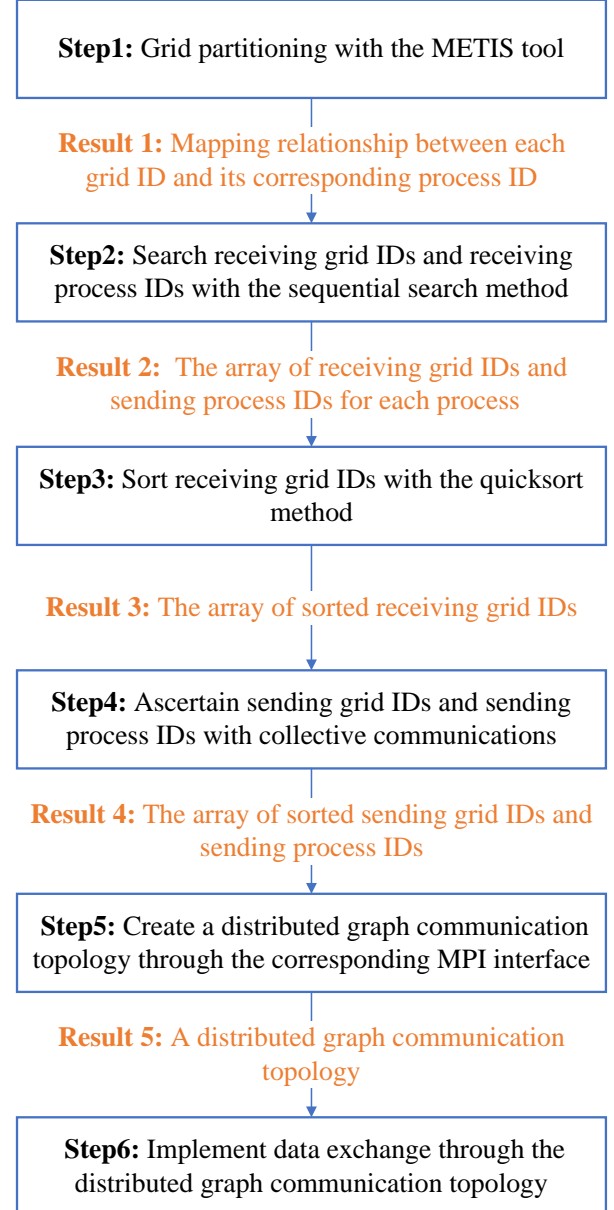

**Figure 4.** Implement data exchange with the distributed graph communication topology

operation is completed. An alternative is to call the non-blocking communication interface MPI_Isend for sending data, but it is infrequently utilized due to the increased complexity that introduces to the sending operation.



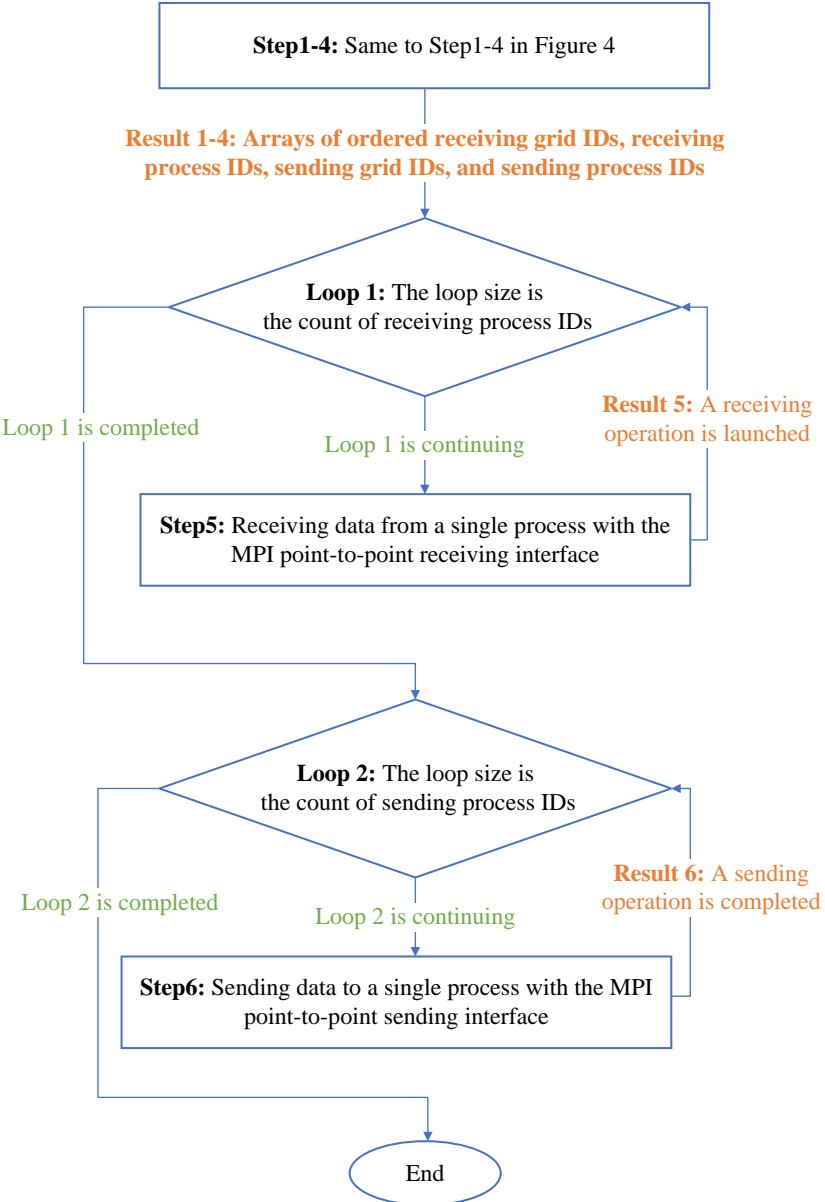

**Figure 5.** Implement data exchange with the point-to-point communication method



**Table 1.** Software and hardware environment

| Name | Version |
| --- | --- |
| CPU | Intel(R) Xeon(R) Gold 6258R CPU 2.70GHz |
| Memory | 196GB |
| Hardware Architecture | X86_64 |
| Operating System | Linux 3.10.0 |
| Compiler | Mpiifort version 2021.6.0 |
| Compiling Option | -O3 |
| MPI | Intel(R) MPI Library 2021.6.0 |

## 4 Experiment

### 4.1 Experiment environment and configuration

The experiments were conducted on the cluster of the National Supercomputing Center of China in Jinan. The software and hardware environment in the test is presented in Table 1.

In the FVWAM configuration, the grid resolution is global $1/12°$, the count of horizontal grids is 6160386, the count of the directional spectrum is 36, and the count of the frequency spectrum is 35. The execution time was calculated by calling the Fortran intrinsic function of *system_clock*. The parameter of *system_clock* is defined as a double-precision integer, and its time-counting frequency is $10^6$ per second.

We performed a series of tests on the FVWAM using different numbers of computing processes, ranging from 512 to 32,768, to evaluate and compare the efficiency between the point-to-point communication method and the distributed graph communication topology. Regarding the distributed graph communication topology, two distinct modes were evaluated: one with reordered processes and another without reordered processes.

### 4.2 Communication performance results

The time step of iterative computation in the test was 60 seconds, and the forecasting period was one hour. Each iteration involved a single neighboring communication for a 3D variable of wave action (*N*). The total times of neighboring communication for *N* during the test was 60. For an equivalent number of processes, the exchanged data and the number of neighboring processes for each process were consistent in both the point-to-point communication method and the distributed graph communication topology. The time of neighboring communication using these two methods is illustrated in Figure 6. The color bar represents the average communication time across all processes, the upper line of the error bar (I) indicates the maximum communication time, and the lower line of the error bar (I) signifies the minimum communication time.

The maximum communication time is a critical factor influencing the performance of the FVWAM model. In comparison to the maximum communication time, the distributed graph communication topology with reordered processes and without





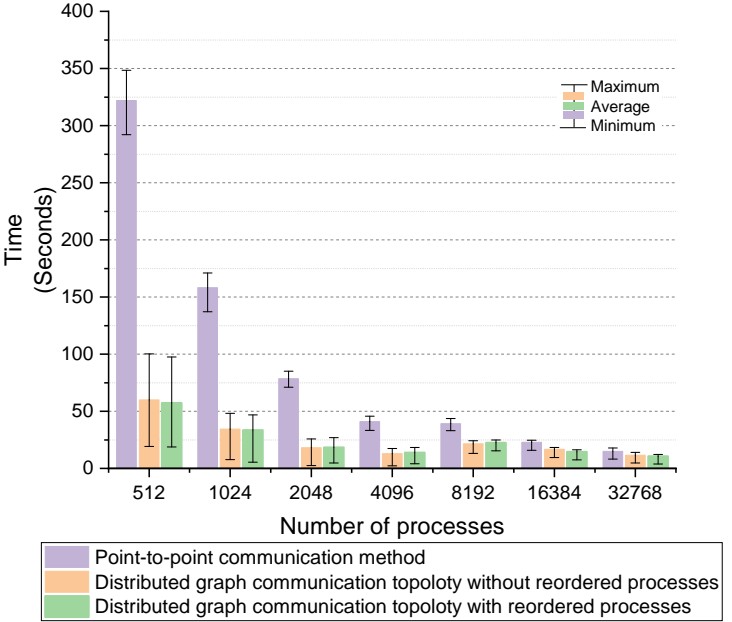

**Figure 6.** Time of neighborhood communication

reordered processes, achieved better performance than the point-to-point communication method, particularly when the count of processes is low. For instance, at a scale of 512 processes, the point-to-point communication method required 348.37 seconds for the maximum communication. In contrast, the distributed graph communication topology with reordered processes took 97.41 seconds, and the distributed graph communication topology without reordered processes exhibited 100.24 seconds.

Regarding the average communication time and the minimum communication time, the distributed graph communication topology with reordered processes and without reordered processes also demonstrated higher communication bandwidth than the point-to-point communication method. The test results indicate that the distributed graph communication topology can improve communication performance, compared to the point-to-point communication method which is prevalent in earth models.

In comparison of the distributed graph communication topology with reordered processes and without reordered processes,
both modes exhibited similar performance levels. However, in the majority of test cases, the performance of the communication topology with reordered processes was slightly better than the communication topology without reordered processes. For instance, with 512, 1024, 8192, 16384, and 32768 processes, the distributed graph communication topology with reordered processes required less average and maximum communication time than the topology without reordered processes. Conversely, with 2048 and 4096 processes, the distributed graph communication topology with reordered processes took more average and
maximum communication time than the topology without reordered processes. The results demonstrate that the distributed graph communication topology with reordered processes does not consistently improve communication performance compared to the topology without reordered processes.





The maximum and average communication times are more critical factors than the minimum communication time for assessing the performance of various communication methods. Based on the maximum and average communication times of the point-to-point communication method, we calculated the speedup ratio for the distributed graph communication topology in Figure 7. Regarding the average time speedup ratio, the performance gap between the distributed graph communication topology and the point-to-point communication method narrows with an increasing number of processes. For instance, at 512 processes, the average time speedup between the distributed graph communication topology with reordered processes and the point-to-point communication method is the highest recorded in the test, yielding a value of 5.63. Two key reasons contribute to this result: First, as the number of processes increases, the volume of exchanged data decreases, thereby reducing the speedup ratio achieved by the distributed graph communication topology. Second, received data are continuously searched and inserted into wave action ($N$) at once in the distributed graph communication topology, which can improve cache hit rates. In contrast, received data are processed separately at times of the point-to-point receiving operations in the point-to-point communication method.

The trend for the maximum time speedup ratio is similar to that of the average time speedup ratio, except at 1024 processes, where the maximum time speedup ratio is marginally higher than at 512 processes. The count of neighboring communication processes on one process may expand as the number of processes rises. This potential increase in communication overhead could explain the improved speedup ratio at 1024 processes compared to 512 processes.

Throughout all tests with 512 to 32,768 processes, compared to the point-to-point method, the time speedup observed with ordered processes and without ordered processes ranged from 1.28 to 5.63. The results substantiate that the distributed graph communication topology can significantly enhance communication efficiency. This improvement is particularly notable when the volume of communication data is high and the number of computing processes is relatively small.

## 4.3 Performance evaluation of the FVWAM

The FVWAM was evaluated over a one-hour forecasting period, during which we measured the communication and computation time excluding the initialization operations and I/O costs. The maximum communication and computation time across all processes is shown in Figure 8(a), and the average communication and computation time is depicted in Figure 8(b). A proportional reduction in computation time was observed with the increase in the number of processes. The computation times using both the distributed graph communication topology and the point-to-point method were nearly identical at the same number of computing processes, indicating that the computing resources utilized in the tests remained stable.

The communication time with the point-to-point communication method decreased as the process count rose from 512 to 32,768, but it declined slower than the computation time. The communication time with the distributed graph communication topology reduced from 512 to 4096 processes, but exhibited fluctuations from 4096 to 32,768 processes. These fluctuations may be attributed to an imbalance in computing load, resulting in fluctuated waiting time during communication. The average communication of the point-to-point communication method accounted for 52.17% to 79.3% of the runtime, with the number of computing processes increasing from 512 to 32,768. In contrast, the distributed graph communication topology with reordered




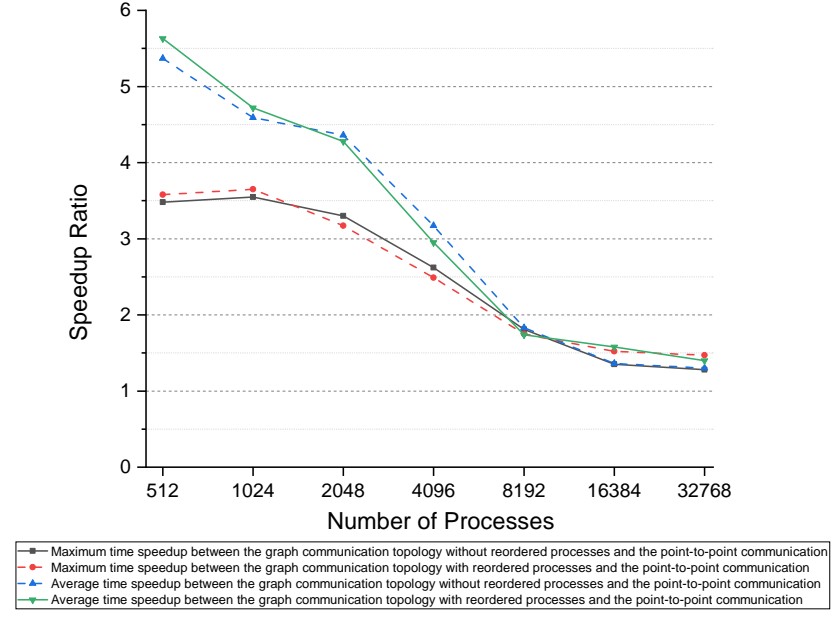

**Figure 7.** Speedup ratio of neighborhood communication

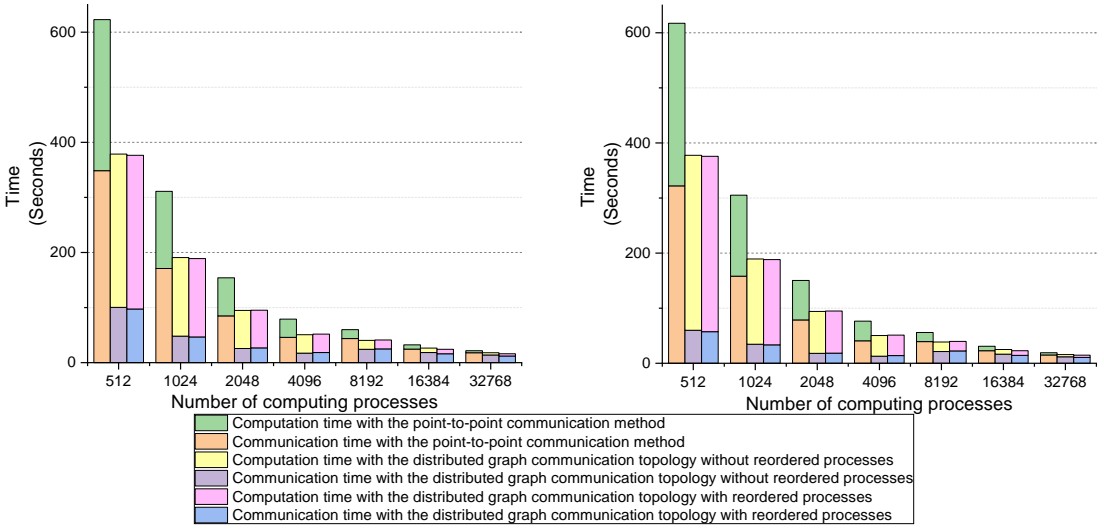

**Figure 8.** (a) Maximum communication and computation time ⠀⠀⠀⠀⠀⠀⠀ (b) Average communication and computation time

processes consumed 15.23% to 72.47% of the runtime over the same range of computing processes. These results underscore that minimizing communication costs is critical to enhancing the scalability of parallel computing models.



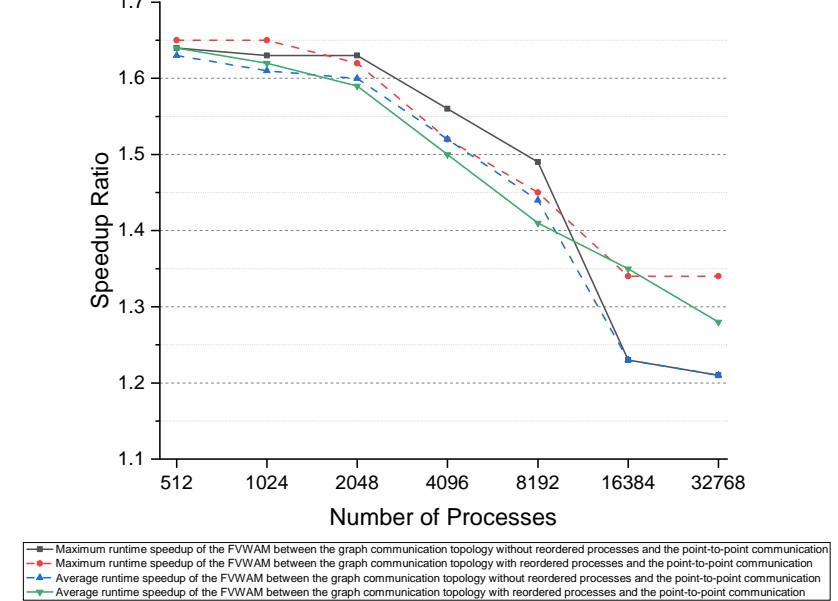

**Figure 9.** Speedup ratio of the FVWAM with various communication methods

We employed the runtime of the FVWAM with the point-to-point communication method as the baseline to calculate the speedup of the model with the distributed graph communication topology, as illustrated in Figure 9. The maximum runtime refers to the actual runtime of the FVWAM excluding I/O and initialization operations. Compared to the point-to-point communication method, the speedup ratios for both the maximum and average runtimes of the FVWAM with the distributed graph communication topology demonstrate a decreasing trend from 1.65 to 1.21. At 512 and 1,024 processes, the speedup ratio for the maximum runtime was 1.65 when contrasting the distributed graph communication topology with reordered processes and the point-to-point communication method. In comparison to the point-to-point method, the minimal speedup ratio for the maximum runtime of the distributed graph communication topology without reordered processes was 1.21 at 32,768 processes in all tests. This indicates that the distributed graph communication topology both with reordered processes and without reordered processes can improve the performance of the model compared to the point-to-point communication method.

In the daily operational context of the FVWAM, 1,024 processes are utilized as the standard computational scale by the NMEFC. By adopting the distributed graph communication topology with reordered processes at this operational computing scale, there is a potential reduction in the iterative runtime of the FVWAM by 40.2% compared to the point-to-point method. This constitutes a substantial enhancement for the operational global wave forecasting service provided by the FVWAM at the NMEFC.



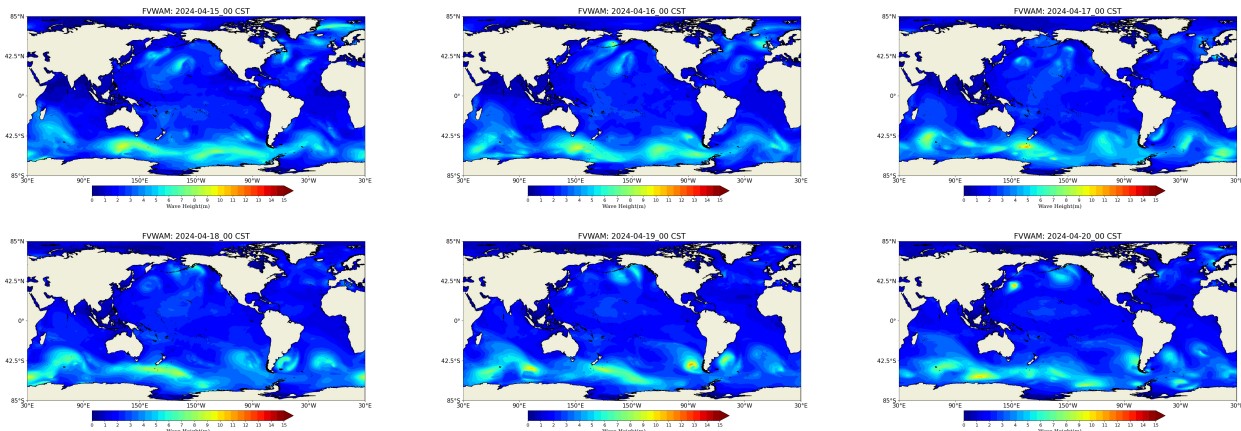

**Figure 10.** Significant wave height products of the FVWAM

## 4.4 Operational products of the FVWAM

The FVWAM with a global resolution of 1/12° is utilized to daily produce the 7-day forecasts for the significant wave height,
wave period, and wave direction at the NMEFC. Among these products, significant wave height is deemed the most crucial
for the mitigation of wave-related disasters. Figure 10 illustrates the actual operational forecast products for significant wave
height at 00:00 UTC+8 from April 15 to April 20, 2024. The measurement unit for significant wave height is meters. Fore-
casters disseminate these FVWAM products along with recommendations for wave disaster mitigation to the public, maritime
transporters, and personnel on ocean platforms to enhance their safety and preparedness.

**5 Conclusions**

In this study, we implemented and compared the point-to-point communication method and the distributed graph communi-
cation topology, utilizing the FVWAM as the case study. The test results led us to conclude that: 1) The distributed graph
communication topology is more efficient than the point-to-point communication method, which is extensively utilized in
earth models, particularly when the number of processes is relatively low. In most cases, the distributed graph communication
topology with reordered processes outperforms the communication topology without reordered processes, although the perfor-
mance gap is modest. 2) Communication cost is a critical factor for the scalability of parallel computing models. Applying the
distributed graph communication topology can significantly enhance the overall performance of the FVWAM, which is crucial
for operational early warning of waves and can be used for communication optimization of other earth models.



*Code and data availability.* The source codes of three versions of the FVWAM using in the case study of this paper are avail-
able at https://github.com/victor-888888/fvwam. The datasets and source codes related to this paper are available via ZENODO at
https://zenodo.org/doi/10.5281/zenodo.13325957.

*Author contributions.* FY, YY planned the project. RP, FY, YG and ZG developed the software design. RP, YG, YY and ZG implemented
the code. RP conducted all performance measurements. RP, YG and LY analyzed the performance measurements. RP, YG wrote the paper,
FY, YY LY and ZG reviewed and revised the paper.

*Competing interests.* The contact author has declared that none of the authors has any competing interests.

*Acknowledgements.* This research was supported by the National Key Research and Development Program of China (Grant No.
2023YFC3107801).



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
