# Peer review of "An Effective Communication Topology for Performance Optimization: A Case Study of the Finite Volume WAve Modeling (FVWAM)"

_EGUsphere, 2024_

## Referee Comment (RC1)

**Review of "An Effective Communication Topology for Performance Optimization: A Case Study of the Finite Volume WAve Modeling (FVWAM)" by Renbo Pang, Fujiang Yu, Yuanyong Gao, Ye Yuan, Liang Yuan, and Zhiyi Gao**

This paper describes implementation and performance benchmarks of neighborhood exchanges in the FVWAM model. It compares the performance of a standard implementation of halo exchanges based on point-to-point communication with the performance of an implementation based on MPI distributed graph topology interface and neighborhood collectives. Using the distributed graph topology interface, the authors obtained a maximum communication time speed-up of 5.63 and 40.2% reduction in the total FVWAM run time with 1,024 processes. This is an interesting and significant result, especially because the use of the distributed graph topology interface and neighborhood collectives is not common in earth system models. I believe the article is suitable for GMD after my comments are addressed.

**Specific comments**

- In section 2.2 the authors describe potential benefits of using the MPI distributed topology interface and present its ability to optimize process mappings as its main advantage. However, the benchmark results of FVWAM show that, while using this interface provides consistent speedups over the point-to-point implementation, setting the reorder flag has only minor performance impacts. What is then the main reason for the observed speedups ? Can section 2.2 be expanded to discuss other potential performance benefits ?

- The presentation of the distributed graph topology workflow in section 3.1 could be improved. The first three paragraphs, related to Figure 3, describe the process of creating MPI graph topology starting from domain partitioning. Most of this material is then repeated in subsequent paragraphs, which pertain to Figure 4. If the authors' intention was to first present the workflow at a high level and then go into details specific to FVWAM this needs to be clearly stated and better organized to remove some of the repetition.

- Are the results of both communication mechanisms bit-for-bit identical ? How was the correctness of the implementation verified ?

- All of the paper performance results were obtained using Intel MPI on one computing system. I imagine that the performance of a high-level interface like the distributed graph topology can strongly depend on the quality of implementation of the underlying MPI library. At minimum, this should

be discussed, but showing results using a different MPI implementation would be a great addition to the paper. Do the authors expect that their results would generalize to other platforms ?

**Minor comments**

- Throughout the paper, the authors refer to cells and cell indices as grids and grid IDs. This terminology is very non-standard and can be confusing. I strongly suggest replacing "grids" with "cells" and either replacing "grid IDs" with "cell IDs" or adding a sentence that in this paper "grid IDs" mean cell IDs.
- Line 156: the variable "cellsOnCell" has already been introduced on line 134, where it is spelled "CellsonCell".
- Line 181: "MPI_DIST_GRAPH_CREATE_ADJACENT" - why are some MPI function names written in all-caps and some are not ? I suggest using the C interface names consistently throughout the paper.
- Lines 212-213: "MPI_Isend (...) is infrequently utilized ...". Can the authors back-up this claim ? All of the models I worked on used "MPI_Isend".
- Table 1: Change "Compiling Option" to "Compilation Options".

---

## Author Response (AR1)

Dear Reviewer,

We would like to sincerely thank you for your thorough and constructive review of our manuscript (An Effective Communication Topology for Performance Optimization: A Case Study of the Finite Volume WAve Modeling (FVWAM)). Your insightful comments have been invaluable in improving the quality of our work. Please find below our detailed responses to each of the comments you raised.

Sincerely,

Renbo PANG, on behalf of the co-authors
* * *
*This paper describes implementation and performance benchmarks of neighborhood exchanges in the FVWAM model. It compares the performance of a standard implementation of halo exchanges based on point-to-point communication with the performance of an implementation based on MPI distributed graph topology interface and neighborhood collectives. Using the distributed graph topology interface, the authors obtained a maximum communication time speed-up of 5.63 and 40.2% reduction in the total FVWAM run time with 1,024 processes. This is an interesting and significant result, especially because the use of the*

*distributed graph topology interface and neighborhood collectives is not common in earth system models. I believe the article is suitable for GMD after my comments are addressed.*

**Reply:** Thank you very much for your positive comments!

***Specific comments 1:*** *In section 2.2 the authors describe potential benefits of using the MPI distributed topology interface and present its ability to optimize process mappings as its main advantage. However, the benchmark results of FVWAM show that, while using this interface provides consistent speedups over the point-to-point implementation, setting the reorder flag has only minor performance impacts. What is then the main reason for the observed speedups? Can section 2.2 be expanded to discuss other potential performance benefits ?*

**Reply :** The original process order is determined by the partitioning scheme of METIS, as described in Lines 150-157. METIS optimizes the process ordering by placing neighboring process IDs together. The MPI implementation then allocates processes across computing nodes according to the ascending order of these process IDs. As a result, the communication performance shows only minor improvements when the reorder flag is set. However, if we were to manually or randomly arrange

the process order instead of using the METIS partitioning result, we believe that the communication performance would improve significantly when the reorder flag is set.

Multiple calls to the point-to-point interface for exchanging data among different processes can lead to unfavorable side effects, such as load imbalance and unpredictable wait times during connection establishment (Torsten et al., 2002). The MPI distributed topology interface optimizes connection management (Torsten et al., 2002). This optimization is independent of whether the reorder flag is set to true or false. We have included this additional information in Section 2.2 to further clarify the benefits of using the MPI distributed topology interface.

**Track changes:** For the comment of "*What is then the main reason for the observed speedups?*", please refer to lines 280-281 and 289-291 in the revised manuscript with track changes.

For the comment of "*Can section 2.2 be expanded to discuss other potential performance benefits?*", please refer to lines 128-131 in the revised manuscript with track changes.

***Specific comments 2:*** *The presentation of the distributed graph topology workflow in section 3.1 could be improved. The first three paragraphs,*

*related to Figure 3, describe the process of creating MPI graph topology starting from domain partitioning. Most of this material is then repeated in subsequent paragraphs, which pertain to Figure 4. If the authors' intention was to first present the workflow at a high level and then go into details specific to FVWAM this needs to be clearly stated and better organized to remove some of the repetition.*

**Reply**:Thank you for your valuable recommendation to improve Section 3.1! The first three paragraphs (Lines 128-148), which relate to Figure 3, have been summarized at a higher level as follows:

The workflow to create a distributed graph communication topology based on SCVT cells is shown in Figure 3. Initially, the global SCVT cells are partitioned according to the number of computing processes. A simple partitioning result of the global SCVT sea cells into three partitions is illustrated in Figure 3(a), with each partition colored green, blue, and purple, respectively.

Next, each process determines its receiving processes and cells based on the partitioning result in Figure 3(a) and the neighboring cell and process information in Figure 3(b). The red line denotes the boundary separating the cells allocated to Processes $P_0$ and $P_1$, while the orange line delineates

the communication boundary for Process $P_0$. The cells situated between the red local cell boundary line and the orange communication boundary line comprise the receiving cells for Process $P_0$.

Finally, a distributed graph communication topology is created by calling the MPI interface with the sending and receiving process IDs and their respective degrees, as shown in Figure 3(c). The sending degree corresponds to the total number of sending processes, and the receiving degree represents the total number of receiving processes.

[Figure]

Figure 3. The workflow to create a distributed graph communication topology

**Track changes:** Please refer to lines 139-161 in the revised manuscript with track changes.

*Specific comments 3: Are the results of both communication mechanisms bit-for-bit identical ? How was the correctness of the implementation verified ?*

**Reply:** Yes, the results of both communication mechanisms are bit-for-bit identical. To verify this, we compared key variables of significant wave height, wave period, and wave direction in the output files generated by both mechanisms. No differences were observed, confirming the correctness of the implementation. We have included this verification of correctness between the point-to-point communication interface and the distributed graph topology interface.

**Track changes:** Please refer to lines 341-344 and 455-458 in the revised manuscript with track changes.

*Specific comments 4: All of the paper performance results were obtained using Intel MPI on one computing system. I imagine that the performance*

*of a high-level interface like the distributed graph topology can strongly depend on the quality of implementation of the underlying MPI library. At minimum, this should be discussed, but showing results using a different MPI implementation would be a great addition to the paper. Do the authors expect that their results would generalize to other platforms ?*

**Reply** : Thank you for your valuable recommendation to compare different communication methods across multiple MPI libraries. Due to the expiration of our rental contract for the high-performance computing system at the National Supercomputing Center of China in Jinan, we are currently unable to conduct additional experiments at the same scale (32,678 CPU cores) with different MPI implementations. However, we conducted smaller-scale tests in the West Pacific region using both Intel MPI Library and Open MPI Library on a different platform. The results indicate that the performance of the distributed graph topology is indeed strongly dependent on the quality of the underlying MPI library implementation.

The software and hardware environment for the first set of tests is presented in Table 1.

Tab.1 Software and hardware environment

| Name | Version |
|---|---|
| CPU | Intel(R) Xeon(R) E5-2680 v4 @ 2.40GHz (28 cores per node) |
| Memory | 128GB |

| Hardware Architecture | X86_64 |
|---|---|
| Network | Infiniband (100Gb/s) |
| Operating System | Red Hat Enterprise 7.6 |
| Compiler | Ifort 17.0.3 |
| Compilation Options | -O3 |
| MPI | Intel(R) MPI Library 2017.3.191 |
| NetCDF | NetCDF-Fortran 4.5.3 |

The cell resolution is 6-12 km, covering the region from 95° E to 145° E and 0° N to 40° N. The number of horizontal cells is 283,517, the count of the directional spectrum is 36, and the count of the frequency spectrum is 35. The time step of iterative computation in the test was 60 seconds, and the forecasting period was one hour. Each iteration involved a single neighboring communication for a 3D variable of wave action $N$. The total times of neighboring communication for $N$ during the test was 60.

We performed a series of tests on the FVWAM using different numbers of computing processes, ranging from 8 to 512 (28 processes per node), to evaluate and compare the efficiency of the point-to-point communication method versus the distributed graph communication topology in the Intel MPI Library, as shown in Figure 10. For intra-node communication with 8 and 16 processes, the performance of both communication methods was similar. However, for inter-node communication, the distributed graph communication topology significantly outperformed the point-to-point method.

[Figure]

Figure 10. Time of neighborhood communication in the Intel MPI Library

The software and hardware environment for the second set of tests is presented in Table 2.

Tab.2 Software and hardware environment

| Name | Version |
|---|---|
| CPU | Intel(R) Xeon(R) E5-2680 v4 @ 2.40GHz (28 cores per node) |
| Memory | 128GB |
| Hardware Architecture | X86_64 |
| Network | Infiniband (100Gb/s) |
| Operating System | Red Hat Enterprise 7.6 |
| Compiler | GNU Fortran 10.2.0 |
| Compilation Options | -O3 |

| MPI | Open MPI 4.0.5 |
| --- | --- |
| NetCDF | NetCDF-Fortran 4.5.3 |

The model configuration in this test is the same as the first test. The results of the FVWAM using different numbers of computing processes, ranging from 8 to 512 (28 processes per node), are shown in Figure 11 to evaluate and compare the efficiency of the point-to-point communication method versus the distributed graph communication topology in the Open MPI Library. The performance gap between the two methods was smaller, and there was no noticeable performance improvement in intra-node communication (with 8 or 16 processes) when using the Open MPI Library, compared to the Intel MPI Library.

[Figure]

Figure 11. Time of neighborhood communication in the Open MPI Library

**Track changes:** Please refer to lines 237-354 in the revised manuscript with track changes.

***Minor comments 1:*** *Throughout the paper, the authors refer to cells and cell indices as grids and grid IDs. This terminology is very non-standard and can be confusing. I strongly suggest replacing "grids" with "cells" and either replacing "grid IDs" with "cell IDs" or adding a sentence that in this paper "grid IDs" mean cell IDs.*

**Reply:** We have replaced "grid" with "cell" which can be confusing , and "grid IDs" have been changed to "cell IDs" throughout the paper.

**Track changes:** Please refer to lines 66, 70, 75, 95, 104, 167, etc.

*Minor comments 2:* *Line 156: the variable "cellsOnCell" has already been introduced on line 134, where it is spelled "CellsonCell".*

**Reply:** The repeated introduction of the variable "cellsOnCell" on line 134 has been removed.

**Track changes:** Please refer to lines 147 in the revised manuscript with track changes.

*Minor comments 3:* *Line 181: "MPI_DIST_GRAPH_CREATE_ADJACENT" - why are some MPI function names written in all-caps and some are not ? I suggest using the C interface names consistently throughout the paper.*

**Reply:** We have replaced "MPI_DIST_GRAPH_CREATE_ADJACENT" with "MPI_Dist_graph_create_adjacent" to ensure consistency with the C

interface naming convention. All MPI function names in the paper have been updated to use the correct C interface names.

**Track changes:** Please refer to lines 195 and 203 in the revised manuscript with track changes.

*Minor comments 4: Lines 212-213: "MPI_Isend (. . . ) is infrequently utilized . . . ". Can the authors back-up this claim ? All of the models I worked on used "MPI_Isend".*

**Reply:** "MPI_Isend (. . . ) is infrequently utilized . . . " on    Lines 212-213 has been removed.

**Track changes:** Please refer to lines 233-234 in the revised manuscript with track changes.

*Minor comments 5:Table 1: Change "Compiling Option" to "Compilation Options".*

**Reply:** The term "Compiling Option" has been changed to "Compilation Options" in the paper.

**Track changes:** Please refer to lines 249-250 in the revised manuscript with track changes.

**Reference:**

Hoefler T, Rabenseifner R, Ritzdorf H, et al. The scalable process topology interface of MPI 2.2[J]. Concurrency and Computation: Practice and Experience, 2011, 23(4): 293-310.

Dear Reviewer,

We would like to sincerely thank you for your thorough and constructive review of our manuscript (An Effective Communication Topology for Performance Optimization: A Case Study of the Finite Volume WAve Modeling (FVWAM)). Your insightful comments have been invaluable in improving the quality of our work. Please find below our detailed responses to each of the comments you raised.

Sincerely,

Renbo PANG, on behalf of the co-authors
* * *
*Paper Review: "An Effective Communication Topology for Performance Optimization: A Case Study of the Finite Volume WAve Modeling (FVWAM)"*

*This paper presents an implementation of halo-exchanges in the FVWAM model using MPI's distributed graph topology and performance comparison over baseline implementation using point-to-point communication primitives.*

*The paper provides a detailed comparison from tests with 512 to 32,768 processes and shows that the speedup from the distributed graph topology with and without reordered processes ranged from 1.28 to 5.63.*

**Reply:** Thank you very much for your valuable and insightful comments!

*There is some important context missing from the article that can shed light on the significance of the performance improvements.*
*\* What is the network interconnect and topology of the target system?*

**Reply:** The network topology used in the tests is a three-layer fat-tree topology, with the primary data exchange network connected by InfiniBand devices. The link bandwidth is 100 Gb/s.

**Track changes:** Please refer to lines 249-250 and 362-363 in the revised manuscript with track changes.

*\* Were experiments repeated with different node allocations assuming there is a batch system scheduling resources?*

**Reply:** The computing nodes are allocated by the Slurm job scheduling system. In all tests, we specified only the number of CPU cores, while the

allocation of nodes was determined by the Slurm. Each experiment was repeated twice, and the better result was selected. This means that the Slurm may allocate different nodes depending on resource availability at the time of execution.

**Track changes:** Please refer to lines 278 in the revised manuscript with track changes.

*\* Were the different experiment types (point-point, distributed, distributed with reordering) conducted using the same node allocation for consistency?*

**Reply:** For the different experiment types with the same number of processes, we did not specify the same node allocation. Node allocation was determined by the Slurm. Since the nodes are shared by multiple user-submitted jobs, specifying the same node list for different experiment types would result in longer wait times, especially for large-scale tests involving up to 32,768 cores.

**Track changes:** Please refer to lines 278 in the revised manuscript with track changes.

*\* Were the experiments at each processor count (e.g., 512) conducted multiple times to rule out network variability and interference from other traffic on the network?*

**Reply:** We repeated the experiments at each processor count twice and selected the better performance result. No significant differences were observed between the repeated tests.

**Track changes:** Please refer to lines 257-259 and 362-364 in the revised manuscript with track changes.

*\* Is there any performance variability across runs?*

**Reply:**Except potentially different computing nodes for different experiment types on the same process count and competition for network bandwidth among different jobs submitted by multiple users mentioned by you in previous comments, I/O operations are also a performance variability. Both the distributed graph communication topology and the point-to-point communication method use the blocking communication mode. As a result, communication time includes the waiting time for receiving data from other processes that have not yet sent data to the current process. Furthermore, I/O imbalances and competition in the global file system among different jobs can lead to varying wait times.

**Track changes:** Please refer to lines 257-259 and 362-364 in the revised manuscript with track changes.

*\* Can the authors elaborate differences of their approach if any with using the MPI-3 neighbourhood collectives?*

**Reply:** MPI-3 offers two methods for neighborhood collectives: Cartesian topology and graph topology. The approach presented in this paper is same with the graph topology in MPI-3, which supports irregular grids and user-defined neighborhood communication. In contrast, the Cartesian topology only supports regular grids, and the number of sources and destinations is fixed at 2×ndims (ndims is the number of dimensions in the Cartesian grid).

**Track changes:** The approach presented in this paper is same with the graph topology in MPI-3, so there is no corresponding changes.

*The authors allude to the following factors as the primary contributors to the improvement:*

*> First, as the number of processes increases, the volume of exchanged data decreases, thereby reducing the speedup ratio achieved by the distributed graph communication topology.*

*It appears that the application is network bandwidth bound at low processor counts. It would be very enlightening to provide details of the communication volume and interconnect specifications to confirm if that's the case.*

**Reply:** As suggested, we calculated the minimum, average, and maximum data volume received by each process, as shown in Figure 12(a) (attached in a separate file), for a range of process counts from 512 to 32,768. The data volume is computed using Formula 1(a). $V_i$ represents the data volume for Process $i$, *num_recv$_j$* denotes the number of grids received from Process $j$, *num_fre* is the frequency of the wave spectrum (set to 35 in the test), *num_dir* is the number of directions in the wave spectrum (set to 36 in the test), *len_data* is the length of one single floating point element (4 bytes), and *steps* is the number of iteration time steps (set to 60 in the test). To simplify the representation of $V_i$, the unit of $V_i$ is expressed in megabytes (MB), calculated by dividing by 1024×1024.

$$V_i = num\_recv_j * num\_fre * num\_dir * len\_data * steps/(1024 * 1024) \qquad \text{(Formula 1)}$$

As the number of processes increases, the average data volume received by each process decreases. The data volume per process ranging from 512 to 32,768 processes is shown in Figure 12(b-h) (attached in a separate file). In these figures, the x-axis represents the receiving Process IDs, the y-axis represents the sending Process IDs, and the color indicates the volume of data received by each process from others. The process ordering is determined by the METIS tool. The results show that most process IDs exchanging data are neighbors, which explains why the performance improvement is less significant after enabling the reordered option in the distributed graph communication topology.

**Track changes:** Please refer to lines 261-281 and 364-373 in the revised manuscript with track changes.

> *Second, received data are continuously searched and inserted into wave action (N) at once in the distributed graph communication topology, which can improve cache hit rates.*

*The presumption about improved cache hit rates can be confirmed by obtaining hardware performance counter information. I'm skeptical that cache performance played such a big role. The improvement could better be explained by MPI library implementation ordering the communication operations optimally.*

*The biggest weakness of this work is the limited performance data from just one platform and MPI implementation. It would highly strengthen the work if the performance optimization can be demonstrated on multiple machines with different interconnects and topologies and recent versions of community standard libraries (OpenMPI, MPICH) or recent vendor implementations. It would make the case for neighbourhood collectives for earth system workloads stronger.*

**Reply:** Thank you for your valuable recommendation to compare different communication methods across multiple MPI libraries. Due to the expiration of our rental contract for the high-performance computing system at the National Supercomputing Center of China in Jinan, we are currently unable to conduct additional experiments at the same scale (32,678 CPU cores) with different MPI implementations. However, we conducted smaller-scale tests in the North West Pacific region using both Intel MPI Library and Open MPI Library on a different platform. The results indicate that the performance of the distributed graph topology is indeed strongly dependent on the quality of the underlying MPI library implementation.

The software and hardware environment for the first set of tests is presented in Table 1.

Tab.1 Software and hardware environment

| Name | Version |
|---|---|
| CPU | Intel(R) Xeon(R) E5-2680 v4 @ 2.40GHz (28 cores per node) |
| Memory | 128GB |
| Hardware Architecture | X86_64 |
| Network | Infiniband (100Gb/s) |
| Operating System | Red Hat Enterprise 7.6 |
| Compiler | Ifort 17.0.3 |
| Compilation Options | -O3 |
| MPI | Intel(R) MPI Library 2017.3.191 |
| NetCDF | NetCDF-Fortran 4.5.3 |

The cell resolution is 6-12 km, covering the region from 90° E to 150° E and 5° S to 45° N. The number of horizontal cells is 283,517, the count of the directional spectrum is 36, and the count of the frequency spectrum is 35. The time step of iterative computation in the test was 60 seconds, and the forecasting period was one hour. Each iteration involved a single neighboring communication for a 3D variable of wave action $N$. The total times of neighboring communication for $N$ during the test was 60.

We performed a series of tests on the FVWAM using different numbers of computing processes, ranging from 8 to 512 (28 processes per node), to evaluate and compare the efficiency of the point-to-point communication method versus the distributed graph communication topology with the Intel MPI Library, as shown in Figure 10. For intra-node communication with 8 and 16 processes, the performance of both communication

methods was similar. However, for inter-node communication, the distributed graph communication topology significantly outperformed the point-to-point method.

[Figure]

Figure 10. Time of neighborhood communication with the Intel MPI Library

The software and hardware environment for the second set of tests is presented in Table 2.

Tab.2 Software and hardware environment

| Name | Version |
| --- | --- |
| CPU | Intel(R) Xeon(R) E5-2680 v4 @ 2.40GHz (28 cores per node) |

| Memory | 128GB |
|---|---|
| Hardware Architecture | X86_64 |
| Network | Infiniband (100Gb/s) |
| Operating System | Red Hat Enterprise 7.6 |
| Compiler | GNU Fortran 10.2.0 |
| Compilation Options | -O3 |
| MPI | Open MPI 4.0.5 |
| NetCDF | NetCDF-Fortran 4.5.3 |

The model configuration in this test is the same as the first test. The results of the FVWAM using different numbers of computing processes, ranging from 8 to 512 (28 processes per node), are shown in Figure 11 to evaluate and compare the efficiency of the point-to-point communication method versus the distributed graph communication topology with the Open MPI Library. The performance gap between the two methods was smaller, and there was no noticeable performance improvement in intra-node communication (with 8 or 16 processes) when using the Open MPI Library, compared to the Intel MPI Library.

[Figure]

Figure 11. Time of neighborhood communication with the OpenMPI Library

**Track changes:** Please refer to lines 305-316 in the revised manuscript with track changes.

*There is work illustrating performance improvements from reordering MPI processes taking network topology into account. e.g., https://dl.acm.org/doi/10.1145/2851553.2851575*

*What is the current reordering strategy in case I missed? Did the authors consider any advanced reordering strategies?*

**Reply:** Thank you very much for providing the reference! In both the point-to-point communication method and the distributed graph communication topology without reordered processes, the process order is same. It is based on the output from the METIS partitioning tool, which is widely used in various models, including MPAS, WAVE WATCH III (WW3), and the Finite Volume Coastal Ocean Model (FVCOM). As shown in Figure 12(a-h), the results of the METIS tool are effective, as it places the majority of communicating processes as neighbors. The reordering strategy used in the distributed graph communication topology with reordered processes depends on the implementation details of the MPI library, which remains a black-box to users.

**Track changes:** Please refer to lines 280-281 in the revised manuscript with track changes.

*The paper refers to pre-posting receives using MPI_Irecv. However, they mention*
*> An alternative is to call the non-blocking communication interface MPI_Isend for sending data, but it is infrequently utilized due to the*

*increased complexity that introduces to the sending operation.*

*It's not inherently that complex as a lot of applications use non-blocking sends effectively. I wonder what the performance impact would be if the authors used non-blocking operations.*

**Reply:** In our tests, we use MPI_Isend to send data and MPI_Recv to receive data in the point-to-point communication method. The distributed graph communication topology, which is implemented using the MPI_Neighbor_alltoallv interface, also employs a blocking communication method. The statement "but it is infrequently utilized due to the increased complexity it introduces to the sending operation" is inaccurate, and it has been removed.

The section titled "Section 3.2 Point-to-point communication method" (Lines 199-214) has been revised and replaced with the following text and graph to better introduce the point-to-point communication method used in the test case.

The approach for implementing the point-to-point communication method for neighboring communication is illustrated in Figure 5. The process for determining ordered arrays of receiving grid IDs, receiving

process IDs, sending grid IDs, and sending process IDs follows the same

steps (1-4) as the distributed graph communication topology described in

Figure 4.

[Figure]

Figure 5. Implement data exchange with the point-to-point

communication method

To prevent communication deadlocks, FVWAM initiates non-blocking

sending operations before starting receiving operations in Step 5. Each process performs sending operations to transmit data to the corresponding receiving processes. These sending operations are executed by repeatedly calling the MPI_Isend interface (*sendbuf*, *destination*, ...). The parameter *sendbuf* refers to the data buffer associated with the sending grid IDs from a single process, and the *destination* parameter corresponds to the receiving process ID. The number of calling the MPI_Isend interface depends on the number of sending process IDs for each process. Since MPI_Isend is non-blocking, it returns immediately without waiting for the completion of the send operation.

In Step 6, each process calls the MPI_Recv interface (*recvbuf*, *source*, ...) to receive data from the sending processes. The *recvbuf* parameter is used to store data corresponding to the receiving grid IDs, and the *source* parameter indicates the sending process ID. MPI_Recv is a blocking communication interface that only completes once the receiving operation has finished.

**Track changes:** Please refer to lines 214-235 in the revised manuscript with track changes.

*I was looking forward to the article to hear about novel techniques that could improve communication performance at scale. However, it was slightly disappointing to see that the benefit from the proposed optimization dramatically tapers off as we go from low (512) to high (32756) number of processes.*

*On heterogeneous GPU based supercomputers like the Frontier Exascale system, the number of nodes is relatively low (9,408) due to the fat node architecture compared to CPU based supercomputers like Fugaku (158,976 nodes) out there. In this overall context, the benefit of a communication optimization if more relevant at scale when there are potentially hundreds of thousands of MPI endpoints at scale (e.g., 600k on Fugaku with 4 MPI ranks per node mapping optimally to the NUMA domains there).*

*Ref: https://docs.olcf.ornl.gov/systems/frontier_user_guide.html*

*https://www.fujitsu.com/global/about/innovation/fugaku/specifications/*
*Pg 15, lines 295-302:*

*To conclude, I understand the motivation of authors to improve their production simulations performance and the relative significance for their workload. Additional performance data would be highly informative and make this more generally applicable.*

**Reply:** Thank you once again for your insightful and constructive comments! As mentioned in our previous response, we conducted additional tests, and the communication times for the three methods are presented in Figures 10 and 11.

Using the maximum and average communication times for the point-to-point communication method with the Intel MPI Library (Figure 10), we computed the speedup ratio for the distributed graph communication topology, as shown in Figure 13. We observed a significant performance gap between intra-node and inter-node communication for the point-to-point communication method. This resulted in similar speedup ratios for both communication methods in intra-node communication with 8 and 16 processes. However, starting from 32 processes, the speedup ratio increases as inter-node communication was introduced.

[Figure]

Figure 13. Speedup ratio of neighborhood communication with the Intel
MPI Library

using the maximum and average communication times for the
point-to-point method with the OpenMPI Library in Figure 11, we
calculated the speedup ratio for the distributed graph communication
topology, which is presented in Figure 14. The results show that the
performance of both communication methods is comparable, indicating
that the performance gap between the two methods depends on the MPI
implementation library used.

[Figure]

Figure 14. Speedup ratio of neighborhood communication with the OpenMPI Library

**Track changes:** Please refer to lines 239-354 in the revised manuscript with track changes.

*Minor comments:*

*Pg 2, line 29: There are better references than Sukhija et al., 2022 for the Frontier Exascale supercomputer. I suggest using one of the papers from the Supercomputing conference. https://dl.acm.org/doi/abs/10.1145/3581784.3607089*

> *Sukhija, N., Bautista, E., Butz, D., and Whitney, C.: Towards anomaly detection for monitoring power consumption in HPC facilities, in: 380 Proceedings of the 14th International Conference on Management of Digital EcoSystems, pp. 1–8, 2022*

**Reply:** As suggested, we have replaced the reference to (Sukhija et al., 2022) with (Atchley et al., 2023)

Atchley, S., Zimmer, C., Lange, J., Bernholdt, D., Melesse Vergara, V., Beck, T., Brim, M., Budiardja, R., Chandrasekaran, S., Eisenbach, M., et al.: Frontier: exploring exascale, in: Proceedings of the International Conference for High Performance Computing, Networking, Storage and Analysis, pp. 1–16, 2023.

**Track changes:** Please refer to line 32 in the revised manuscript with track changes.

*The performance sections in the paper are a bit verbose and redundant pointing to the information in the figures. It might be better to be succinct in highlighting the results and elaborate further on the reasons behind the improvement.*

**Reply:** We have revised this section to make it more concise, focusing on directly highlighting the key results rather than reiterating the details already presented in the figures. We have also expanded on the underlying reasons behind the observed improvements to provide a clearer understanding of the factors contributing to the performance gains. These revisions have be included in the revised manuscript submission.

**Track changes:** Please refer to lines 379-414 and 437-441 in the revised manuscript with track changes.

---

## Referee Report (RR1)

**Review of revised version of "An Effective Communication Topology for Performance Optimization: A Case Study of the Finite Volume WAve Modeling (FVWAM)" by Renbo Pang, Fujiang Yu, Yuanyong Gao, Ye Yuan, Liang Yuan, and Zhiyi Gao**

The revised paper is greatly improved. I especially welcome the addition of Open MPI results. I believe the article is nearly ready for publication in GMD. I have just a few minor comments.

**Minor comments**

- Following my suggestion, the authors renamed grid IDs to cell IDs in the text. However, figures 4 and 5 still refer to grid IDs. This needs to be fixed.

- In Section 4.1 the authors show new performance results of halo exchange with Intel MPI and Open MPI in small-scale parallel experiments. They compare the graph-based and point-to-point implementations for each MPI library. They conclude that there is a large speed-up from using the distributed graph topology only with the Intel library. However, I don't see any discussion of the fact that the point-to-point method using Open MPI is significantly faster than the same method with Intel MPI for inter-node communication. This suggests that the large speed-up from using the graph implementation with the Intel library is partly because the point-to-point implementation is performing poorly. I think this should be mentioned in the paper.

- The added plots of communication data volume are interesting, but, for me at least, there are too many of them. I don't think it is necessary to show the process versus process plot for every process count. Most of them are very similar and don't add much. It would be sufficient to show them only for the smallest and the largest number of processes.

- In Table 2 it would be good to include compilation options again. Otherwise, it looks like only the Intel runs used compiler optimization. Also, why is the version of NetCDF included in Table 1 ? I don't see how it is relevant, considering the paper doesn't discuss I/O performance.

---

## Author Response (AR2)

Dear Reviewer,

We would like to sincerely thank you for your second thorough and constructive review of our manuscript (An Effective Communication Topology for Performance Optimization: A Case Study of the Finite Volume WAve Modeling (FVWAM)). Your insightful comments have been invaluable in improving the quality of our work. Please find below our detailed responses to each of the comments you raised.

Sincerely,

Renbo PANG, on behalf of the co-authors
* * *
*The revised paper is greatly improved. I especially welcome the addition of Open MPI results. I believe the article is nearly ready for publication in GMD. I have just a few minor comments.*

**Reply**:Thank you very much for your positive comments!

***Minor comments 1:*** *Following my suggestion, the authors renamed grid IDs to cell IDs in the text. However, figures 4 and 5 still refer to grid IDs. This needs to be fixed.*

**Reply**:The "grid IDs" in Figures 4 and 5 have been replaced by "cell IDs".

**Track changes:** Please refer to Figures 4 and 5 between Lines 208 and 209 in the revised manuscript with tracked changes.

*Minor comments 2: In Section 4.1 the authors show new performance results of halo exchange with Intel MPI and Open MPI in small-scale parallel experiments. They compare the graph-based and point-to-point implementations for each MPI library. They conclude that there is a large speed-up from using the distributed graph topology only with the Intel library. However, I don't see any discussion of the fact that the point-to-point method using Open MPI is significantly faster than the same method with Intel MPI for inter-node communication. This suggests that the large speed-up from using the graph implementation with the Intel library is partly because the point-to-point implementation is performing poorly. I think this should be mentioned in the paper.*

**Reply**:Discussion between Open MPI and the Intel MPI have been added as following in the manuscript. Different MPI implementations may utilize varying communication patterns, process binding and scheduling

strategies, buffer management techniques, and support for remote direct memory access (RDMA). These differences can lead to performance disparities among MPI implementations. For intra-node communication, both the distributed graph communication topology and the point-to-point communication method using the Intel MPI library significantly outperformed their counterparts implemented with the Open MPI library. Liu et al. (2003) stated that the Intel MPI library enhances performance by leveraging shared memory mechanisms for intra-node communication. For inter-node communication, the distributed graph communication topology exhibited similar performance with both the Intel MPI and Open MPI libraries. However, the point-to-point communication method with the Open MPI library demonstrated significantly superior performance compared to the Intel MPI library. Rashti (2010) identified high overhead for small messages with the Intel MPI library, which was attributed to semantics miscorrelation between MPI and the user-level library. A further potential cause could be the lack of RDMA support or poor performance with buffer management for small messages in the point-to-point communication method using the Intel MPI library.

**Track changes:** Please refer to lines 286-297 in the revised manuscript with track changes.

*Minor comments 3: The added plots of communication data volume are interesting, but, for me at least, there are too many of them. I don't think it is necessary to show the process versus process plot for every process count. Most of them are very similar and don't add much. It would be sufficient to show them only for the smallest and the largest number of processes.*

**Reply:** Only the smallest, middle (for aesthetic purposes to maintain an even number of images), and largest numbers of processes were retained; the others have been removed.

**Track changes:** Please refer to Figure 6 between Lines 262 and 263, and Figure 7 in Page 26 in the revised manuscript with track changes.

*Minor comments 4: In Table 2 it would be good to include compilation options again. Otherwise, it looks like only the Intel runs used compiler optimization. Also, why is the version of NetCDF included in Table 1 ? I don't see how it is relevant, considering the paper doesn't discuss I/O performance.*

**Reply**:As suggested, the version of NetCDF has been removed.

**Track changes:** Please refer to Table 1 in Page 11 in the revised manuscript with track changes.

**Reference:**

Rashti, M. J.: Improving Message-Passing Performance and Scalability in High-Performance Clusters, Ph.D. thesis, Queen's University, 2010.

Liu, J., Chandrasekaran, B., Wu, J., Jiang, W., Kini, S., Yu, W., Buntinas, D., Wyckoff, P., and Panda, D. K.: Performance comparison of MPI implementations over InfiniBand, Myrinet and Quadrics, in: Proceedings of the 2003 ACM/IEEE conference on Supercomputing, p. 58, 2003.

---

## Author Response (AR3)

Dear Katja Gänger,

We would like to sincerely thank you for your constructive review of our manuscript (An Effective Communication Topology for Performance Optimization: A Case Study of the Finite Volume WAve Modeling (FVWAM)). Your insightful comments have been invaluable in improving the quality of our work. Please find below our detailed responses to each of the comments you raised.

Sincerely,

Renbo PANG, on behalf of the co-authors
* * *
***Minor comments 1:*** *a) Regarding figures 2, 3, 6, and 14: the figures' panels are included separately with their own captions. This will be unacceptable in the final paper.*

**Reply**:The subfigure captions in Figures 2, 3, 6, and 14 have been removed.

**Track changes:** Please refer to Figure 2 between Lines 121 and 122, Figure 3 between Lines 166 and 167, Figure 6 between Lines 261 and

262, and Figure 14 on page 26 in the revised manuscript with tracked changes.

***Minor comments 2:*** *b) FYI: The supplement zip file was removed because it contains the source files of your manuscript which will be requested at a later stage.*

**Reply:Thank you very much for your kind reminder.**